# Efficient Local Unlearning for Gaussian Processes with Out-of-Distribution Data

**Juliusz Ziomek**
University of Oxford
{firstname}@robots.ox.ac.uk

**Ilija Bogunovic**
University College London

## Abstract

Gaussian Processes (GPs) offer robust uncertainty estimates crucial for data-efficient applications like Black-box Optimization or Model Predictive Control. However, when the underlying function changes, previously gathered data can mislead predictions, impacting performance. Instead of indiscriminately removing all data points (or a large fraction) after detecting a change, the goal is to efficiently identify and remove only the obsolete data points, a process we refer to as unlearning in GPs. Leveraging the model's uncertainty estimates, we transform the unlearning problem into one of maximizing variance (nearly reverting to GP prior values) at detected change points by selectively removing the most informative training points. Though the exact solution to this problem is NP-hard, we propose an efficient algorithm that approximates the optimal solution while significantly reducing computational complexity. This algorithm utilizes novel fast *reverse update* equations for GP models, enabling linear-time sequential computation of the posterior variance function with removed training points.We test the performance of our unlearning procedure across various tasks, including Model Predictive Control, Transfer Bayesian Optimization, and Time-Varying Bayesian Optimization. Our approach offers a comprehensive solution for handling out-of-distribution issues in GP modeling, *significantly* outperforming baseline methods.

## 1 Introduction

Gaussian Processes (GPs) [16] are a class of models used to estimate a function over a domain of interest based on limited observations of that function's values. Because of good uncertainty estimates, GPs found applications in domains, where the agent needs to be data-efficient and actively query for new data points, such as Black-box Optimisation or Model Predictive Control. However, if the target function is non-stationary or undergoes a one-time change, some previously gathered data may become obsolete. This out-of-distribution data can mislead the model, resulting in incorrect predictions and reduced task performance. One naive solution to this problem is to remove all data points once we realize the function we wish to model has changed. However, this approach can be very wasteful, especially if the change occurs in only a small domain region, leaving much of the training data still valid. Instead, by observing discrepancies between model predictions and actual observed values, we can pinpoint the obsolete data points that influenced those erroneous predictions and remove them from the training set. Given that GPs provide uncertainty estimates, this problem can be reframed as removing training points to maximize the GP variance at the points where we detected function changes. We refer to this process as *local unlearning* in GPs and address this specific problem setting.

Identifying a small number of data points to remove is, unfortunately, an NP-hard problem [7]. It requires considering all possible combinations of training points to remove in order to find the one that maximizes the variance at desired points. Combined with the cubic complexity of fitting a GP

Workshop on Bayesian Decision-making and Uncertainty, 38th Conference on Neural Information Processing Systems (NeurIPS 2024).

model, this makes the exact solution computationally infeasible. However, in this paper, we present an *efficient* algorithm that approximates the solution in a fraction of the time required for an exact combinatorial search. We derive *fast reverse update equations* for GPs, allowing us to compute the variance function with a given training point removed at a computational cost that is only linear in the number of training points. We provide a performance guarantee for this algorithm and state its time complexity. We then demonstrate how the algorithm can address distribution shifts in Model Predictive Control tasks, Transfer Bayesian Optimization, and Time-Varying Bayesian Optimization.

**Related Work:** The problem of transferring data between tasks using Gaussian Processes has also been extensively explored in the context of Multi-Task GPs [3, 9, 13] and transfer Bayesian Optimization (BO) [14, 17]. These studies typically assume that the training tasks share similarities with the target task. Our work tackles an issue in which some previously gathered data may be misspecified, shifting our focus from data transfer to effectively performing unlearning.

Efficient forward posterior updates in Gaussian Processes are commonly employed, especially in sequential Bayesian optimization (see, e.g., Appendix F in [4]). However, to the best of our knowledge, there is limited understanding regarding efficient reverse updates in GPs (i.e., computing posterior updates after removing observations). [11] introduced a method for computing the downgraded Cholesky matrix that scales quadratically with the number of observations. While downgrading the Cholesky matrix allows for predictive variance calculations at any point, our equations provide a time complexity advantage when predictive variance is needed only at a limited number of fixed points.

## 2 Efficient unlearning

Assume a problem setting when we use the GP model with the posterior predictive variance function defined as $\sigma_{\mathcal{D}}^2(x) = k(x,x) - \boldsymbol{k}_{\mathcal{D}}^T (\boldsymbol{K}_{\mathcal{D}} + \sigma^2 I_{|\mathcal{D}|})^{-1} \boldsymbol{k}_{\mathcal{D}}$, where $\boldsymbol{k}_{\mathcal{D}} \in \mathbb{R}^T$ with elements $(\boldsymbol{k}_{\mathcal{D}})_i = k(x,x_i)$ and $\boldsymbol{K}_{\mathcal{D}} \in \mathbb{R}^{T \times T}$ with entries $(\boldsymbol{K}_{\mathcal{D}})_{i,j} = k(x_i,x_j)$ . Let $\mathcal{U}$ be a set of points in a input domain $\mathcal{X}$ for which we detected anomalies. We thus aim for our uncertainty estimate at those points to revert (almost) to its prior state, as though no learning had occurred. However, we note that for typically used kernels, such as RBF or Matérn, for any two points in the input domain $x, x' \in \mathcal{X}$ we have that $k(x,x') > 0$. As a result, if we want to revert variance exactly to the prior, we need to remove all of the training points. As such, we instead wish to be $\eta$-close to the prior variance. We define this concept formally below.

**Definition 2.1.** *For a specified unlearning set $\mathcal{U} \subset \mathcal{X}$ and unlearning precision $\eta > 0$, let $\sigma_{\mathcal{D}}^2(x)$ represent the model's current estimate based on the complete dataset $\mathcal{D}$. The goal of unlearning generating is to select a set of points subset $\mathcal{S} \subset \mathcal{D}$ to remove to produce a new estimate of posterior variance $\sigma_{\mathcal{D} \setminus \mathcal{S}}^2(x)$, such that:*

$$\min_{\mathcal{S} \subset \mathcal{D}} |\mathcal{S}| \quad s.t. \ \forall u \in \mathcal{U} \ \sigma_{\mathcal{D} \setminus \mathcal{S}}^2(u) \geq \sigma_{\emptyset}^2(u) - \eta,$$

*where $\sigma_{\emptyset}^2(x)$ corresponds to the prior GP variance.*

Note that the optimisation constraint can also be written in an equivalent form: $g_{\mathcal{D}}(\mathcal{S}) = g_{\mathcal{D},\max}$, where the following function

$$g_{\mathcal{D}}(\mathcal{S}) = \sum_{u \in \mathcal{U}} \left( \min\{\sigma_{\mathcal{D} \setminus \mathcal{S}}^2(u), \sigma_{\emptyset}^2(u) - \eta\} - \min\{\sigma_{\mathcal{D}}^2(u), \sigma_{\emptyset}^2(u) - \eta\} \right) \tag{1}$$

is the *variance gain* function and $g_{\mathcal{D},\max} = \sum_{u \in \mathcal{U}} \max\{\sigma_{\emptyset}^2(u) - \eta - \sigma_{\mathcal{D}}^2(u), 0\}$ is the maximum variance gain. Notice that we can think about $g_{\mathcal{D},\max}$ as a type of loss function. The quantity $g_{\mathcal{D} \setminus \mathcal{S},\max}$ tells us how what is the maximum variance gain we can still achieve after we have already removed a set of points $\mathcal{S}$. Ideally, after we remove the chosen set of points, we would like this quantity to be as small as possible. However, solving this problem is, in general, NP-hard, which means the exact solution requires combinatorial time complexity. We will show how the solution to this problem can be efficiently approximated with a greedy algorithm. Such a greedy algorithm at each timestep will remove the point $\boldsymbol{x}$ that produces greatest variance increase, i.e. $\max_{x \in \mathcal{D} \setminus \mathcal{S}} g_{\mathcal{D} \setminus \mathcal{S}}(\{x\})$. However, to compute $g_{\mathcal{D} \setminus \mathcal{S}}(\{x\})$ efficiently, we need to derive fast downgrade equations, which we do next.

Assuming we have access to the model matrix $\Delta^{\mathcal{D}} = (K_{\mathcal{D}} + \sigma^2 I_{|\mathcal{D}|})^{-1}$ for datapoints in $\mathcal{D}$, we will now show how one can quickly compute the variance function for the model with a datapoint

removed. We also show how the model matrix $\Delta^{\mathcal{D}}$ itself can be efficiently downgraded, once we decide on the point we want to remove at a given iteration. Without the loss of generality, we will assume the point we want to remove is the last point with an index of $T$. We now present equations allowing for fast computations of reverse updates in a GP.

$$\sigma^2_{\mathcal{D}\setminus\{x_T\}}(u) = \sigma^2_{\mathcal{D}}(u) + \frac{1}{\Delta^{\mathcal{D}}_{T,T}}(\mathbf{k}^T_{\mathcal{D}}\Delta^{\mathcal{D}}_{T,1:T})^2 \tag{2}$$

$$\Delta^{\mathcal{D}\setminus\{x_T\}} = \Delta^{\mathcal{D}}_{1:T-1,1:T-1} - \frac{\Delta^{\mathcal{D}}_{1:T-1,T}\Delta^{\mathcal{D}}_{T,1:T-1}}{\Delta^{\mathcal{D}}_{T,T}}, \tag{3}$$

where $\Delta^{\mathcal{D}}_{T,1:T}$ is the $T$th column of $\Delta^{\mathcal{D}}$ corresponding to point $x_T$ and $\Delta^{\mathcal{D}}_{T,T}$ is the $T$th diagonal entry in $\Delta^{\mathcal{D}}$. We derive those equations in Appendix B. We now propose our efficient GP unlearning procedure in Algorithm 1. At each iteration of the `while` loop, the Algorithm (greedily) removes the point that produces the highest increase in variance. To find that point, the Algorithm iterates through all points by the `for` loop in lines 4-7. In line 5, the Algorithm uses fast reverse updates for the GP to measure the variance value at a given point in $\mathcal{U}$ given that the point in question is removed.

---

**Algorithm 1** Efficient unlearning algorithm

---

**Require:** unlearning set $\mathcal{U}$, Training Data Points $\mathcal{D}$, Unlearning precision $\eta$, Stopping criterion $\gamma$, Mean and Variance of Original Model $\mu_{\mathcal{D}}(\cdot)$, $\sigma^2_{\mathcal{D}}(\cdot)$, Inverted Model Matrix $\Delta^t$
1: Initialise set for removed points $\mathcal{S} = \emptyset$
2: **while** $g_{\mathcal{D}\setminus\mathcal{S},\max} > \gamma g_{\mathcal{D},\max}$ **do**
3:     For each $u \in \mathcal{U}$ compute $\sigma^2_{\mathcal{D}\setminus\mathcal{S}}(x) = k(x,x) - \boldsymbol{k}^T_{\mathcal{D}\setminus\mathcal{S}}(\boldsymbol{K}_{\mathcal{D}\setminus\mathcal{S}} + \sigma^2 I_{|\mathcal{D}\setminus\mathcal{S}|})^{-1}\boldsymbol{k}_{\mathcal{D}\setminus\mathcal{S}}$
4:     **for** each datapoint $x \in \mathcal{D}\setminus\mathcal{S}$ **do**
5:         For each $u \in \mathcal{U}$ compute $\sigma^2_{\mathcal{D}\setminus(\mathcal{S}\cup\{x\})}(u)$ given $\sigma^2_{\mathcal{D}\setminus\mathcal{S}}(u)$ by using Equation 2
6:         Use $\sigma^2_{\mathcal{D}\setminus\mathcal{S}}(u)$ to compute $g_{\mathcal{D}\setminus\mathcal{S}}(\{x\})$ and store it
7:     **end for**
8:     Find the next point to remove $x = \arg\max_{x\in\mathcal{D}} g_{\mathcal{D}\setminus\mathcal{S}}(\{x\})$
9:     Compute new inverted model matrix $\Delta^{\mathcal{D}\setminus(\mathcal{S}\cup\{x\})}$ given the old $\Delta^{\mathcal{D}\setminus\mathcal{S}}$ using Equation 3
10:    Compute $g_{\mathcal{D}\setminus(\mathcal{S}\cup\{x\}),\max} = g_{\mathcal{D}\setminus\mathcal{S},\max} - g_{\mathcal{D}\setminus\mathcal{S}}(\{x\})$
11:    Add this point to the set of removed points $\mathcal{S} := \mathcal{S} \cup \{x\}$
12: **end while**

---

In Appendix D we show that the running time of Algorithm 1 is $(|\mathcal{D}|^4|\mathcal{U}||\mathcal{S}^\star|\log\frac{1}{\gamma})$. Additionally we have the following result (which we prove in Appendix C) on the performance of the algorithm.

**Theorem 2.2.** *Assume the variance decrease function $F_x(\mathcal{D}) := \sigma^2_\emptyset(x) - \sigma^2_s(x)$ is submodular. Let $\mathcal{S}_{greedy}$ be the set of points removed by the greedy algorithm until a $\gamma$-approximation to the optimal solution can be found, that is $g_{\mathcal{D}\setminus\mathcal{S}_{greedy},max} \leq \gamma g_{\mathcal{D}\setminus\mathcal{S}^\star,max}$. We then have that:*

$$|\mathcal{S}_{greedy}| \leq |\mathcal{S}^\star|\log\frac{1}{\gamma},$$

*where $\mathcal{S}^\star$ is the optimal solution to problem statement in Definition 2.1.*

The assumption of submodularity of $F_x(\mathcal{D})$ is standard [2] and prior work established conditions that guarantee it[5]. As such, we see that a greedy algorithm can achieve $1 - \gamma$ of the maximum possible variance gain while removing only $\log\frac{1}{\gamma}$ more points than the exact algorithm. We now proceed to show how our algorithm can be applied in practical problem settings.

## 3 Experiments

We implemented our experiments in Python. We share our code via the following *anonymised* link[1]. See details of each experiment in Appendix A. On each experiment, we classify an observation as an anomaly if it lies outside of the 95% confidence interval.

---

[1] `https://github.com/JuliuszZiomek/EfficientGPUnlearning`

## 3.1 Time varying Bayesian Optimisation

We perform time-varying Bayesian Optimisation on the Intel Research Dataset [2]. The dataset consists of temperature recordings gathered over 50 sensors placed in the Intel office in Berkeley. Our goal is to select the sensor with the highest temperature at each timestep and the regret is the difference between the highest and selected temperature. We show results in Figure 2. The baselines, we compared against are keeping all points (Keep All) and sliding window (SW) algorithms [18] with windows sizes of 5 and 10. We can see that after around iteration 40 all other methods start to underperform compared to unlearning and at around iteration 80 they suffer a drastic jump in the average regret values. Inspecting the number of points kept by unlearning at each iteration, we see that our algorithm removes almost all datapoints around iteration 40 and after that almost all new points are kept. This would imply that the underlying function experienced a change around that iteration and after that remained relatively stable. The SW algorithms appear to be suffering from catastrophic forgetting, whereas the baseline keeping all of the points is using obsolete data from before the function change.

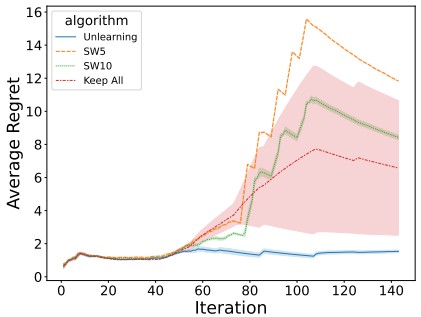 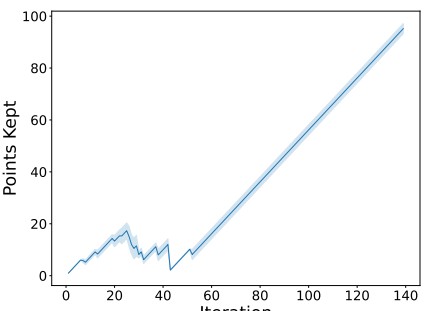

Figure 1: Results on the Time-Varying BO experiments on the Intel Reserach Dataset. Left subplot shows average regret at each iteration for all algorithms and the right plot shows the number of points that the unlearning algorithm kept (i.e. have not removed) against the iteration. Shaded areas are standard erros over six seeds.

## 3.2 Transfer Bayesian Optimisation

Next, we consider the problem of transferring data from one BO problem to another, we describe the details in Appendix A. We show the results in Figure 2 As baselines, we use the TAF method of [17], as well as the two state-of-the-art Multi-Task GP baselines proposed by [14], namely WSGP and SHGP. We can see that utilising prior data gives a head start to the method, compared to optimising the hyperparameters from scratch. However, the method that simply keeps all data points quickly gets stuck at a sub-optimal solution and eventually gets outperformed by the optimiser that does not use prior data. On the other hand, our unlearning algorithm is able to converge to the same optimum as the freshly initialised optimiser, but much faster. Inspecting the number of points kept by the unlearning algorithm, we see that it removes most of the points except for 10-20, which seem crucial to the completion of the task. TAF method is unable to converge within the number of iterations tried, whereas WSGP and SHGP converge to similar solutions as unlearning, but at a slower pace.

## 3.3 Model Predictive Control with Domain Shifts

We consider two control problems that experience a domain shift. The first one is a modification of the cart pole problem, where after a number of iterations half of the ground becomes frozen, increasing the breaking distance. The second problem is rusty pendulum, where at some point the bearings become rusty and the maximum torque that can be applied on one side of the pendulum is reduced. We show the average returns on both problems after the change occurred in Figure 3a

---

[2]https://db.csail.mit.edu/labdata/labdata.html

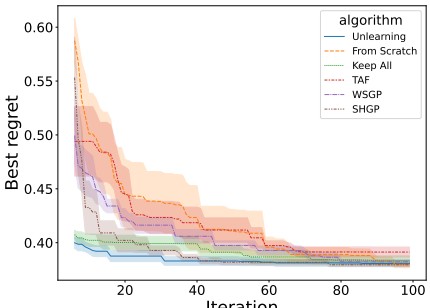 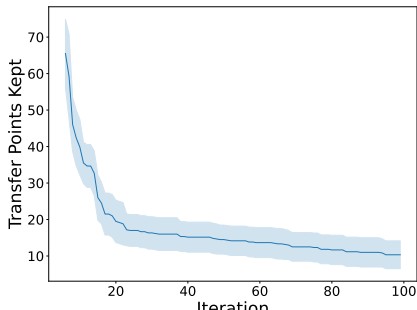

Figure 2: Results of the Transfer BO experiment on wine datasets. The plot on the left shows the best regret achieved by each method, whereas the plot on the right shows the number of points that were kept (i.e. not removed) by our unlearning algorithm. Shaded areas are standard errors over six seeds.

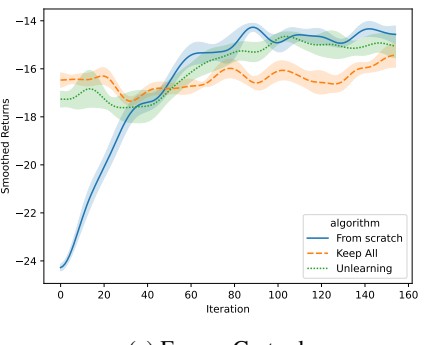

(a) Frozen Cartpole

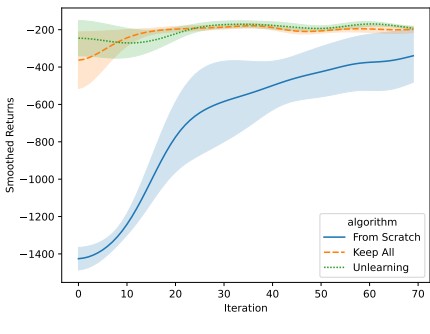

(b) Rusty Pendulum

Figure 3: Results on a Model Predictive Control problems. The results are over 5 seeds and shaded areas correspond to standard errors. The plots are displaying smoothed returns, where the smoothing has been done by a gaussian filter with $\sigma = 5$.

and 3b. Two baselines we compare against are keeping all data points (Keep All) and removing all data after the change has occurred (From Scratch). On the Frozen Cartpole problem, we see that the unlearning algorithm does not get stuck at the suboptimal return values, unlike the algorithm keeping all points. At the same time, in early iterations, its return is much higher than the algorithm learning from scratch. On the Rusty Pendulum problem, within the training episodes, the unlearning algorithm reaches optimal return faster than learning from scratch and is more stable than the strategy keeping all points.

## 4    Conclusions

Within this paper, we addressed the important problem of local unlearning in GP, necessitated by domain shifts in the underlying function we wish to model. We developed an efficient approximation to the otherwise computationally expensive problem and showed how it can be applied to a number of real-world problems involving GPs. One limitation of our work is that we only considered standard GP models. When dealing with large of datapoints, a typical practice is to switch to sparse GP [12]. Extending our framework to deal with sparse models constitutes an exciting direction for future work.

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

# A  Experiments details

## A.1  Compute Resources

To run all experiments we used a machine with AMD Ryzen Threadripper 3990X 64-Core Processor and 252 GB of RAM. No GPU was needed to run the experiments. We were running multiple runs in parallel. One run for BO experiments took up to five minutes, whereas the runs for MPC took up to two hours.

## A.2  Time-Varying BO experiments details

We used the data from the 1st and 2nd March to compute covariance between sensors and use it as the kernel for GP. Then we run optimisation on all of the data gathered on the 3rd of March. We treat one timestep as 10 minutes and thus the temperature of a given sensor at each timestep is its average temperature in this interval.

## A.3  MetaBO experiment details

We consider the problem of fitting a random forest regressor to the famous wine dataset [1] to predict the quality of wine expressed as a number from 1 to 10 based on different features of the wine. The problem here is to find five hyperparameters of the model (two of which are combinatorial), to minimise validation set mean square error. We first run BO for 100 steps on the red wine dataset. We then wish to solve the same problem for the white wine dataset, with the hope that some of the data from the red wine case could be reused. For this experiment, we use a GP with Transformed Overlap [15] kernel and utilise the MCBO[6] implementation. For unlearning, we set $\eta = 0.3$ and $\gamma = 0.5$.

## A.4  Model predictive control details

We describe the environments below. To run the experiments, we use the codebase provided by [10] [3].

### Frozen Cartpole

We consider a modification of the famous cart pole control problem, where having previously gathered data on the standard cart pole problem, we realise that part of the ground has frozen. As such, the dynamics of the system are different there, because the breaking distance is longer. We first gather data on the unmodified cartpole problem by running standard MPC for 300 iterations, before running it in the modified domain. For the unlearning algorithm, we select $\eta = 0.7$ and $\gamma = 0.5$ and limit the number of points to remove to 150.

### Rusty Pendulum

As a second problem, we consider the pendulum benchmark, where one needs to apply torque to a pendulum so as to stabilise it in an upright position. We modify a problem setting, where after some time, the right part of the pendulum's bearing becomes rusty and thus the resulting friction on that side is greater. As such, the maximum torque we can apply in rusted places is reduced and the optimal strategy is now to try to reach an upright position by rotating the pendulum through the left side of the bearings. We first gather data from 200 iterations on the standard pendulum problem. For the unlearning algorithm, we select $\eta = 0.7$ and $\gamma = 0.5$ and limit the number of points to remove to 25.

---

[3]`https://github.com/fusion-ml/trajectory-information-rl`

## B Derivation of Fast Update Equations

Assuming we know matrix $\Delta \in \mathbb{R}^{T \times T}$, for which $\Delta = (K_T + \sigma^2 I_T)^{-1}$ and we wish to learn $(K_{T-1} + \sigma^2 I_{T-1})^{-1}$. Using Schur's complement we get

$$
\Delta = \begin{pmatrix} A & B \\ C & D \end{pmatrix}^{-1} = \begin{pmatrix} A^{-1} + A^{-1}B\beta C A^{-1} & -A^{-1}B\beta \\ -\beta C A^{-1} & \beta \end{pmatrix}
$$
$$
= \begin{pmatrix} A^{-1} + \beta A^{-1}BB^T A^{-1} & -\beta A^{-1}B \\ -\beta B^T A^{-1} & \beta \end{pmatrix}
$$
$$
= \begin{pmatrix} A^{-1} + \beta\gamma & -\beta\alpha \\ -\beta\alpha^T & \beta \end{pmatrix},
$$

where $\beta = (D - CA^{-1}B)^{-1} = 1/(D - B^T A^{-1}B)$, $\gamma = A^{-1}BB^T A^{-1}$, and $\alpha = A^{-1}B$. Notice that $A^{-1} = (K_{T-1} + \sigma^2 I_{T-1})^{-1}$. If we want to know $A^{-1}$ from $\Delta$, we can compute it using the following equation:

$$
A^{-1} = \Delta_{1:T-1,1:T-1} - \beta\gamma
$$
$$
= \Delta_{1:T-1,1:T-1} - \beta\alpha\alpha^T
$$
$$
= \Delta_{1:T-1,1:T-1} - \frac{(-\beta\alpha)(-\beta\alpha^T)}{\beta}
$$
$$
= \Delta_{1:T-1,1:T-1} - \frac{\Delta_{1:T-1,T}\Delta_{t,1:T-1}}{\Delta_{t,T}},
$$

which can be computed in $\mathcal{O}(T^2)$. If we want to get $\mu_{T-1}$ and $\sigma^2_{T-1}$, we can compute them as follows:

$$
\mu_{T-1}(x) = \mathbf{k}_{T-1}^T A^{-1} \mathbf{y}_{T-1}
$$
$$
= \mathbf{k}_{T-1}^T \Delta_{1:T-1,1:T-1}\mathbf{y}_{T-1} - \mathbf{k}_{T-1}^T \frac{\Delta_{1:T-1,T}\Delta_{t,1:T-1}}{\Delta_{T,T}}\mathbf{y}_{T-1}
$$
$$
= \sum_{i=1}^{T-1}\sum_{j=1}^{T-1} k_i \Delta_{i,j} y_j - \frac{1}{\Delta_{t,T}}\sum_{i=1}^{T-1} k_i \Delta_{i,T} \sum_{j=1}^{T-1} y_j \Delta_{t,j}
$$
$$
= \sum_{i=1}^{T}\sum_{j=1}^{T} k_i \Delta_{i,j} y_j - \sum_{j=1}^{T-1} k_T \Delta_{t,j} y_j - \sum_{i=1}^{T-1} k_i \Delta_{i,T}\mathbf{y}_T - k_T \Delta_{t,T}\mathbf{y}_T - \frac{1}{\Delta_{t,T}}\sum_{i=1}^{T-1} k_i \Delta_{i,T}\sum_{j=1}^{T-1} y_j \Delta_{t,j}
$$
$$
= \sum_{i=1}^{T}\sum_{j=1}^{T} k_i \Delta_{i,j} y_j - \frac{1}{\Delta_{t,T}}\sum_{i=1}^{T} k_i \Delta_{i,T}\sum_{j=1}^{T} y_j \Delta_{T,j}
$$
$$
= \mathbf{k}_T^T \Delta \mathbf{y}_T - \frac{1}{\Delta_{T,T}}(\mathbf{k}_T \Delta_{1:t,T})(\Delta_{T,1:T}\mathbf{y}_T)
$$
$$
= \mu_T(x) - \frac{1}{\Delta_{T,T}}(\mathbf{k}_T \Delta_{1:t,T})(\Delta_{T,1:T}\mathbf{y}_T)
$$
$$
\sigma^2_{T-1}(x) = k(x,x) - \mathbf{k}_{T-1}^T A^{-1}\mathbf{k}_{T-1}
$$
$$
= k(x,x) - \mathbf{k}_{T-1}^T \Delta_{1:T-1,1:T-1}\mathbf{k}_{T-1} + \mathbf{k}_{T-1}^T \frac{\Delta_{1:T-1,T}\Delta_{T,1:T-1}}{\Delta_{T,T}}\mathbf{k}_{T-1}
$$
$$
= k(x,x) - \mathbf{k}_T^T \Delta \mathbf{k}_T + \mathbf{k}_T^T \frac{\Delta_{1:t,T}\Delta_{T,1:T}}{\Delta_{T,T}}\mathbf{k}_T
$$
$$
= \sigma^2_T(x) + \frac{1}{\Delta_{T,T}}(\mathbf{k}_T^T \Delta_{1:t,T})(\Delta_{T,1:T}\mathbf{k}_T) = \sigma^2_T(x) + \frac{1}{\Delta_{T,T}}(\mathbf{k}_T^T \Delta_{1:t,T})^2,
$$

where each can be computed in $\mathcal{O}(T)$.

# C Proof of Theorem 2.2

*Proof.* Let us define the the following function $G_{x,\mathcal{D}}(S') := -F(\mathcal{D}/S') = \sigma^2_{\mathcal{D}\setminus S'}(x) - \sigma^2_\emptyset(x)$. Consider sets $Y \subset S$ and $Y' \subset S'$ and some $x \in \mathcal{D}$. We then have that:

$$
\begin{aligned}
G(S' \cup \{x\}) - G(S') &= F(\mathcal{D} \setminus S') - F(\mathcal{D} \setminus (S' \cup \{x\})) \\
&= F(S \cup \{x\}) - F(S) \\
&\leq F(Y \cup \{x\}) - F(Y) \\
&= G(\mathcal{D} \setminus Y) - G(\mathcal{D} \setminus (Y \cup \{x\})) \\
&= G(Y' \cup \{x\}) - G(Y'),
\end{aligned}
$$

where the inequality comes from the assumption on submodularity of $F(\cdot)$. The property we show above is one of the equivalent conditions for submodularity of $G(S')$, which shows $G(\cdot)$ is also submodular.

Notice the following:

$$
\begin{aligned}
g_\mathcal{D}(\mathcal{S}) &= \sum_{u \in \mathcal{U}} \left( \min\{\sigma^2_{\mathcal{D}\setminus\mathcal{S}}(u), \sigma^2_\emptyset(u) - \eta\} - \min\{\sigma^2_\mathcal{D}(u), \sigma^2_\emptyset(u) - \eta\} \right) \\
&= \sum_{u \in \mathcal{U}} \left( \sigma^2_\emptyset(u) + \min\{\sigma^2_{\mathcal{D}\setminus\mathcal{S}}(u) - \sigma^2_\emptyset(u), -\eta\} - \sigma^2_\emptyset(u) - \min\{\sigma^2_\mathcal{D}(u) - \sigma^2_\emptyset(u), -\eta\} \right) \\
&= \sum_{u \in \mathcal{U}} \left( \min\{G_{x,\mathcal{D}}(S), -\eta\} - \min\{G_{x,\mathcal{D}}(\emptyset), -\eta\} \right).
\end{aligned}
$$

Since $G_{x,\mathcal{D}}(S)$ is a submodular function of $\mathcal{S}$ and taking the minimum and summation preserves submodularity, the function $g_\mathcal{D}(\mathcal{S})$ is also submodular. We further notice the following property:

$$
\begin{aligned}
g_{\mathcal{D}\setminus\mathcal{S},\max} - g_{\mathcal{D}\setminus\mathcal{S}}(\{x\}) &= \sum_{u \in \mathcal{U}} \max\{\sigma^2_\emptyset(u) - \eta, \sigma^2_{\mathcal{D}\setminus\mathcal{S}}(u)\} - \sigma^2_{\mathcal{D}\setminus\mathcal{S}}(u) - g_{\mathcal{D}\setminus\mathcal{S}}(\{x\}) \\
&= \sum_{u \in \mathcal{U}} \sigma^2_\emptyset(u) - \eta - \min\{\sigma^2_{\mathcal{D}\setminus(\mathcal{S}\cup\{x\})}(u), \sigma^2_\emptyset(u) - \eta\} \\
&= \sum_{u \in \mathcal{U}} \max\{\sigma^2_\emptyset(u) - \eta - \sigma^2_{\mathcal{D}\setminus(\mathcal{S}\cup\{x\})}(u), 0\}) \\
&= g_{\mathcal{D}\setminus(\mathcal{S}\cup\{x\})}(u)
\end{aligned}
$$

Following the proof idea of Theorem 3.1 of [2], we can rely on Lemma 2 of [8] to get that:

$$
|\mathcal{S}_{\text{greedy}}| \leq |\mathcal{S}^*| \log \frac{1}{\gamma} \quad \text{to achieve} \quad g_{\mathcal{D}\setminus\mathcal{S}_{\text{greedy}},\max} \leq \gamma g_{\mathcal{D}\setminus\mathcal{S}^\star,\max}
$$

$\square$

# D Runtime complexity of Algorithm 1

Due to Theorem 2.2, we have that the `while` loop will be executed at most $|\mathcal{S}^\star| \log \frac{1}{\gamma}$ times. Line 3, requires variance computations for each of the points in $\mathcal{U}$ and as such incurs the complexity of $\mathcal{O}(|\mathcal{D}|^2|\mathcal{U}|)$. Each time the `for` loop in lines 4-7 is executed, we need to compute the variance with a point removed for each $u \in \mathcal{U}$, which can be done in $\mathcal{O}(|\mathcal{D}||\mathcal{U}|)$ and as such the complexity of the entire loop is $\mathcal{O}(|\mathcal{D}|^2|\mathcal{U}|)$. The only remaining computationally intensive operation is the computation of the new inverted model matrix in line 9 and this can be done in $\mathcal{O}(|\mathcal{D}|^2)$. This brings the complexity of each `while` loop execution to $\mathcal{O}(|\mathcal{D}|^2|\mathcal{U}|)$ and the complexity of the entire algorithm to $\mathcal{O}(|\mathcal{D}|^2|\mathcal{U}||\mathcal{S}^\star| \log \frac{1}{\gamma})$. Note that this is an improvement from the naive algorithm, which would fit GP every time with one point removed, resulting in complexity $(|\mathcal{D}|^4|\mathcal{U}||\mathcal{S}^\star| \log \frac{1}{\gamma})$.

