# OpenReview forum: "Efficient Local Unlearning for Gaussian Processes with Out-of-Distribution Data"
_NeurIPS.cc/2024/Workshop/BDU — NeurIPS BDU Workshop 2024 Poster_

### Official Review · Reviewer_faYM · 2024-09-26
**The manuscript presents an efficient posterior update algorithm for local unlearning in GPs.**

**Rating:** 8
**Confidence:** 3

**Review:**

The manuscript is well-organized and written to a high standard, with a focus on addressing out-of-distribution issues in GPs, which has a wide range of applications such as transfer learning and Bayesian optimization. The authors formulated an optimization problem to unlearn GPs , developed a linear time GP posterior update method, and implemented the algorithm in many real-world applications to demonstrate its efficiency.

**Questions and Comments:**

1. Are there any theoretical explanations for why the proposed method outperforms others in Figure 1, and how it avoids getting stuck in local optima? Does this advantage stem from maximizing the variance? While the paper emphasizes the computational benefits, it would be interesting to explore how the optimization setup affects solution convergence. So if  the authors can provide a theoretical explanation, it could further strengthen the argument for the proposed method.

2. In Appendix B.2, the authors state, "We used data from the 1st and 2nd of March to compute the covariance between sensors and use it as the kernel for the GP." This is unclear—if the covariance is based on historical data, wouldn’t it be a fixed value? How can it be used as a kernel function in the following computations?

3. Also in Appendix B.2, it is stated, "Instead of tuning $\eta$ and $\gamma$, we found it easier to tune the algorithm by limiting the unlearning procedure to remove only one point per iteration when an anomaly is detected." On what basis was this statement made? Was it made purely for simplicity in the experimental setup?

4. It would be helpful to briefly describe the baseline methods used in the experiment and how they differ from the proposed approach, either in Appendix B (Experiment Details) or the related work section. If readers are unfamiliar with some of the algorithms, it may be difficult for them to interpret the results shown in the experiment. Additionally, the experimental details are somewhat not detailed enough. While it is understandable that some details may be omitted due to the workshop format, providing enough information for readers to reproduce the experiments based on the manuscript would be beneficial.

---

### Official Review · Reviewer_UCVh · 2024-09-28
**A solid work, where reasons to accept outweigh reasons to reject.**

**Rating:** 6
**Confidence:** 4

**Review:**

Local unlearning for Gaussian Processes (GPs) refers to updating or modifying a GP model when encountering out-of-distribution (OOD) data points. Addressing this challenge involves removing all possible combinations of training points to identify the one that maximizes the variance at the desired points, a problem known to be NP-hard. This paper introduces an efficient algorithm that approximates the optimal solution while significantly reducing computational complexity.

The paper is well-structured and represents solid work. Although it mainly proposes a heuristic method, the theoretical analysis is sound. Additionally, it includes a discussion on computational complexity, which is essential for this type of work.

The main concern is that the experimental analysis does not adequately support all of the claims made in the paper. More in-depth numerical discussions are needed, as Figure 1 alone is insufficient to demonstrate the model's superiority.

While the proposed method is more efficient than other baselines, the complexity analysis shows that it still requires considerable time when the training size is large.

---

### Decision · Program_Chairs · 2024-10-09

Accept (Poster)